# Sores of boreal moose reveal a previously unknown genetic lineage of parasitic nematode within the genus *Onchocerca*

**Bridgett M. Benedict**[1]*, **Perry S. Barboza**[1,2], **John A. Crouse**[3], **Katia R. Groch**[4], **Matthew R. Kulpa**[4], **Dan P. Thompson**[3], **Guilherme G. Verocai**[4], **Dominique J. Wiener**[4]

**1** Department of Ecology and Conservation Biology, Texas A&M University, College Station, Texas, United States of America, **2** Department of Rangelands Wildlife and Fisheries Management, Texas A&M University, College Station, Texas, United States of America, **3** Alaska Department of Fish and Game, Division of Wildlife Conservation, Kenai Moose Research Center, Soldotna, Alaska, United States of America, **4** Department of Veterinary Pathobiology, School of Veterinary Medicine & Biomedical Science, Texas A&M University, College Station, Texas, United States of America

* bbenedict@tamu.edu

**Data Availability Statement:** All files are available from the Texas Data Repository database and

## Abstract

Long-standing reports of open sores on the hind legs of moose (*Alces alces*) have been recorded in Alaska (as well as Canada, Europe, and Michigan), eliciting concerns about causes and infection. We used histological and genomic methods to investigate the sores from 20 adult moose on the Kenai Peninsula, Alaska. We paired this with thermal imagery and molt scoring of adult moose to further describe sore formation and understand its timing. Severe, ulcerative and eosinophilic dermatitis was found in all moose with sores present, and microfilariae within intraepidermal pustules were additionally found in four samples. Genetic analysis of sores from moose revealed a previously unknown genetic lineage of *Onchocerca*. Adult moose molt and lose their barrier of protection against flies in June and July during peak fly activity, leaving them vulnerable and allowing the development of sores. In summary, our results indicate that the cause for the sores on the hindleg of moose is a previously unknown genetic lineage of *Onchocerca*, probably transmitted by black flies, in timing with the molt cycle of adult moose. These sores leave moose exposed to pathogens, making them vulnerable, and challenging their health and fitness.

## Introduction

A mammal's coat and integument is an important barrier between the animal and its environment, providing protection against pollutants, extreme temperatures, pathogens, and irritants such as insects [1–3]. Hair and melanin in the integument offer thermoregulation and photoprotection. Any break in the integument is a potential site of infection by a wide variety of pathogens that can cause morbidity and mortality [3]. The ability to repair these breaks can be an indication of the viability of an animal.

Ungulates such as horses (*Equus caballus*), donkeys (*Equus asinus*), mules (*Equus asinus* x *Equus caballus*), and a dromedary camel (*Camelus dromedarius*) often have been documented with round sores/breaks in the integument called "summer sores" (cutaneous habronemiasis)

GenBank: Benedict, Bridgett, 2022, "Moose Sores and Molt", https://doi.org/10.18738/T8/HGPOGJ, Texas Data Repository, V1 GenBank Accession No. OP265723-39.

**Funding:** This work was supported by the Alaska Department of Fish and Game Federal Wildlife Restoration Grant (grant number AKW-4 Project No. 1.63) and the Boone & Crockett Dr. James H. "Red" Duke endowment for Wildlife Conservation and Policy at Texas A&M University.

**Competing interests:** The authors have declared that no competing interests exist.

[3]. These summer sores are caused from a fly (Diptera) bite transferring spirurid nematodes (*Habronema majus*, *H. microstoma*, *H. muscae*, and *Draschia megastoma*), whose larvae erratically migrate through the tissue, initiating an infection, and causing a gross lesion with ulcerated granulated tissue to grow [3]. From the surface, the lesions range from 5–15 cm in diameter and there may be one or many, starting small and irregular and growing to circular sores [3].

Another type of lesion known as "legworm" or "footworm" has been found on the distal legs of moose (*Alces alces*), caribou (*Rangifer tarandus*), white-tailed deer (*Odocoileus virginianus*), mule deer (*Odocoileus hemionus*), elk (*Cervus canadensis*), and pronghorn (*Antilocapra americana*), and has been associated with *Onchocerca cervipedis* [4–6]. *O. cervipedis* is a filarioid nematode that is transmitted through an arthropod vector and, as an adult worm, infects primarily the subcutaneous tissues of ungulate legs and hooves. These adult parasites will produce microfilariae, which will move to the skin tissues throughout the ungulate body, to be taken up by intermediate black fly vectors (Diptera: Simuliidae) [7, 8]. Once ingested, microfilariae will develop to the infective L3 stage and migrate to black fly mouth parts to infect other mammalian hosts upon a subsequent blood meal. These parasite-induced lesions seem to have no seasonal pattern, are often found along the metatarsus or metacarpus, and occur in subcutaneous tissues with gross lesions only visible when the area is skinned [9, 10].

Contrary to the subcutaneous lesions associated with *O. cervipedis*, open sores have been found proximal on the hind leg of moose, on the area above the tibio-tarsal joint, also known as the hock [11, 12]. Up to 12 round sores of approximately 1.5 cm in diameter have been recorded in moose from Alaska, Michigan, Canada, and Europe [12, 13]. Despite a superficial resemblance suggesting a similar cause, these sores have not been reported as cutaneous habronemiasis. The timing of the appearance of these sores in mid-June and declining in early September coincides with the timing of fly activity in boreal areas of North America, suggesting biting flies as the potential cause and the main factor delaying the healing [11–13]. It has been suggested that moose flies (*Haematobosca alcis*) are responsible for creating these sores based on their mouth parts and observations of them feeding on the periphery of the sores, aligning themselves in whorl patterns over the sores [11, 12]. Horse flies (*Hybomitra* spp.) and leeches (Clitellata) have also been suggested, though leeches are the least probable cause as they have rarely been observed [11, 12].

Areas on a moose with a lower density of hairs and shorter hairs are heavily targeted by feeding flies [2, 14, 15]; after molting, the caudal aspect of the legs, the perianal region, and the eyes are most vulnerable to flies [12]. A moose's coat consists of guard hairs covering their entire body, wool hairs or underfur on their torso, and vibrissae around their eyes and nose [16]. Adult moose experience one annual molt beginning in late spring and summer, whereas moose calves molt from neonatal coats to winter coats in the late summer [17, 18].

Very little is known about the open hind leg sores on moose. The principal goal of this project was to describe and analyze these sores, with the main objective being to identify the cause for the lesions. We first describe the progression of molt and the formation of sores in adult moose, using visual scoring and thermal imagery. We then sampled the sores of moose and used histological and genomic techniques to describe the sores and identify the parasitic agent associated with these lesions.

## Materials and methods

### Captive facility

All procedures for care, handling, and experimentation of animals were approved by the Animal Care and Use Committee, Alaska Department of Fish and Game (ADFG), Division of

Wildlife Conservation (IACUC protocol no. 0086-2019-38 and 0086-2020-40) and by the Agricultural Animal Care and Use Committee, Texas A&M AgriLife Research (AUP 2019-009A and 2021-009A).

We studied captive moose held at the Kenai Moose Research Center (MRC), operated by the ADFG on the Kenai National Wildlife Refuge, Alaska, USA (60° N, 150° W). All captive female moose (2–18 y old) used in this study (2015: n = 11; 2016: n = 12; 2021: n = 12) were held in 2.6 km$^2$ outdoor enclosures. Moose had unlimited access to water and natural forage habitats; mixed seral state boreal forest, black spruce (*Picea mariana*) forest, bogs, open meadows, and lakes.

## Molt and hock scoring

We observed all MRC moose at 5–20 day intervals from May 5 to July 13, 2015; May 3 to July 18, 2016; and May 19 to August 13, 2021. During these observations, or from photographs taken during these observations, each individual was assigned a molt score based on their progress in whole body molting (Table 1 and S1 Fig). Individuals were also assigned a hock score to further track the progression from hair coverage to the appearance of sores on their hind legs, in the area above the hock (Table 1 and S2 Fig). Our molt and hock scoring system was designed to best capture the molt sequence that we observed in previous years and during our observations. Adult molt is characterized by the loss of the long guard hairs and underfur and the replacement with shorter guard hairs, which continues to grow into the long guard hairs associated with a winter coat. During 2021 observations, we also recorded the number of sores per individual's leg.

## Hind leg thermal imagery

We took thermal images of the moose from May 19 to August 13, 2021 to track warming of their hind legs across the season. We used a forward-looking infrared thermal camera (FLIR T1030sc; FLIR Systems) with a 12° x 9° lens (f/1.2), and pixel resolution of 1,024 x 768. We stood approximately 5 m away directly facing the rear end of the moose, and captured the area from the rump to the hocks while the moose were standing. These images were taken in the

**Table 1. Molt and hock scores.** Scores and description of each score used to describe the stages of the annual molt. Scores and description of each score used to describe the hair loss and progression towards sore being apparent on the hind leg area above the hocks of the moose.

| Molt Score | Description |
|---|---|
| 1 | None–no signs of molt |
| 2 | Start of Ears–ears starting to molt |
| 3 | Ears Molted–ears molted and/or nose and eyes molting |
| 4 | Loose Body Hair–loose winter hair on body |
| 5 | Around Eyes Molted–molted around eyes and thin winter hair coverage on legs |
| 6 | Thin Shoulders & Legs–face molted from eyes forward, thin winter hair coverage on shoulders and legs |
| 7 | Mostly Molted–mostly short summer body hair, face mostly molted |
| 8 | Molted–full summer coat |
| **Hock Score** | **Description** |
| 1 | Hair–hair covering the area above the hock |
| 2 | Hair Loss–small amounts of hair loss above the hock |
| 3 | No Hair–clear hair loss above the hock |
| 4 | Sores–sores above the hock |

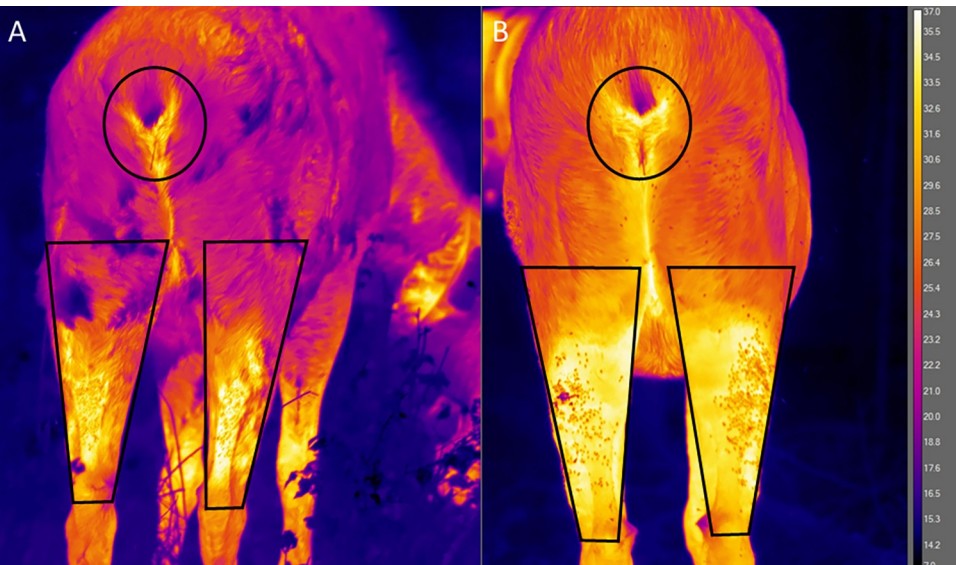

**Fig 1. Thermal analysis.** Surface temperature of the same moose on May 28, 2021 (A) and July 30, 2021 (B). Surface temperatures were measured inside of the ellipses (black circles) and polygons (black irregular quadrilaterals) drawn over the images, representing perianal and leg temperatures respectively.

shade, under clouds or forest canopy. Multiple images were taken and the sharpest image with the best view was used for analysis. Ambient air temperature (˚C) and relative humidity (%) were recorded digitally with a portable weather meter (Kestrel 4400 Heat Stress tracker or Kestrel DROP, Kestrel, Boothwyn, PA, USA) to correspond with each image. We analyzed the images using FLIR ResearchIR Max software (Version 4.40.1; 64 bit, FLIR Systems), and corrected them to an emissivity of 0.95, distance of 2 m, and the correlated ambient air temperature and relative humidity, using spatial calibration calculated for focal length of 83.2 mm with pixel pitch of 17 microns [18, 19]. For each selected image we created three regions of interest (ROI) (Fig 1). First, an ellipse ROI was drawn to encompass the perianal region and vulva of the moose to calculate a reference temperature for the exposed skin. Then, a polygon ROI was drawn on each hind leg encompassing the hind leg area from the center line of the hock to the base of the leg to calculate the maximum and minimum leg temperatures.

We regressed leg and perianal temperature against date and climatic variables (STATA 15.1; StataCorp, College Station, Texas, USA). We used the robust "sandwich estimator" for standard errors to relax assumptions of normal distribution and homogeneity of variances [20]. We used a reverse stepwise selection procedure for all regressions, which removed coefficients that were not significantly different from zero. All statistical significance was set at $P \leq 0.05$. The model for maximum leg temperature (max_temp) included Julian day (day) and ambient air temperature (T) as continuous fixed effects: max_temp = day + T. Similarly, the model for minimum leg temperature (min_temp) included Julian day (day) and ambient air temperature (T) as continuous fixed effects: min_temp = day + T. The model for maximum perianal temperature (perianal) included Julian day (day) and ambient air temperature (T) as continuous fixed effects: perianal = day + T.

## Tissue collection

We immobilized the moose from May 12 to July 22, 2021 with Thiafentanil oxalate (0.001–0.004 mg/kg estimated body mass; 10mg/mL; ZooPharm Wildlife Pharmaceuticals Inc.,

Windsor, CO, USA) and Xylazine (0.03–0.05 mg/kg estimated body mass; 100mg/mL; Lloyd Laboratories, Shenandoah, IA, USA) hand-injected deep into shoulder muscle using a luer-lock syringe and 21Ga x 25mm hypodermic needle. Immobilization lasted less than 45 minutes in duration and was reversed with Atipamezole HCl (0.005 mg/kg estimated body mass; ¼ dose intravenous, ¾ dose intramuscular; 5 mg/mL; Zoetis, Parsippany, NJ, USA) and intra-muscularly administered Naltrexone HCl (100 mg/mg Thiafentanil oxalate intramuscular; 50mg/mL; ZooPharm LLC, Laramie, WY, USA). We monitored the heart, respiration rate and blood perfusion to the mucous membranes of the eyes and gums throughout handling. When hind leg sores were present, we took a biopsy of the sore most proximal on the accessible leg (the other leg was beneath the immobilized moose). If a vessel was directly beneath the sore, the next closest sore was selected. If sores were not present, we took a biopsy in the same area that sores are known to occur later in the year. Scarring and changes in hair density and color were used as indicators for this location. We took the biopsy by gently rotating and pressing a sterile, disposable 6-mm punch (Miltex biopsy Sterile Disposable Dermal Punch, Integra, Princeton, NJ, USA) in the middle of the sore. The punched tissue sample was then removed with forceps and cut at the base with a scalpel blade. The tissue sample was then cut down the center line with a scalpel, half was placed in 2.5 ml RNAlater Stabilizing Solution (Invitrogen, Carlsbad, CA, USA) and half was placed in 2.5 ml 10% buffered formalin. We flushed the wound with isotonic saline (0.9% sodium chloride) and antiseptic cream (4% Chlorhexidine gluconate) was applied to the wound. After 24 hours, we removed the samples in formalin and rinsed them with 70% ethanol and placed in 2.5 ml ethanol. We later shipped the samples to Texas A&M University; prior to shipment we drained the ethanol-soaked samples of ethanol and placed an ethanol-soaked gauze pad for shipment.

## Wild moose sampling

We also collected tissue samples from wild moose that were killed from collisions with vehicles (2020: n = 2 adults; 2021: n = 8 adults, n = 1 calf) on the Kenai Peninsula, Alaska, USA from September 3 to 24, 2020 and from June 30 to July 23, 2021. Excised pieces of skin and muscle from the hock area were collected by ADFG personnel. We trimmed the hair on these samples and cut out the sores and some surrounding tissue using scissors and a scalpel, stopping at muscle ($\leq$ 5mm). Skin without sores was also sampled for comparison. At least one sore from each individual was placed in RNAlater Stabilizing Solution and the remainder were placed in 10% buffered formalin. If only one sore was present, then the sample was cut in half and one side was placed in RNAlater Stabilizing Solution and the other half was placed in 10% buffer formalin. We prepared and shipped the samples as described above.

## Histology

We trimmed the tissue sections and placed them in cassettes for processing. The tissue was embedded in paraffin, cut at 4 μm thick sections and stained with hematoxylin and eosin using the standard procedures. The sections were evaluated by a board-certified veterinary patholo-gist (DJW).

## Genomic DNA

We performed Genomic DNA extraction from RNAlater preserved sections of moose leg tis-sue using DNeasy Blood & Tissue Kits (Qiagen, CA, USA) according to the manufacturer's recommendations. In total 26 samples were processed; 13 samples from dead wild moose and 13 samples from MRC moose. We amplified DNA extracts for the partial cytochrome *c* oxidase subunit 1 (COI) of the mitochondrial DNA (mtDNA) and performed polymerase chain

reaction (PCR) reactions in 25 μL containing 0.25μM of each primer, 1x GoTaq® Green Master Mix (Promega Corporation, Madison, Wisconsin, United States) and 2.5 μL of DNA template. We amplify the COI using primers COINT (forward) 5'-TCAAAATATGCGTTCTACT GCTGTG-3' and COINT (reverse) 5'-CAAAGACCCAGCTAAAACAGGAAC-3' using a protocol modified from Hassan et al. [21]. Briefly, the cycling conditions consisted of an initial denaturation 95˚C for 2 min, followed by 35 cycles of 95˚C for 45 s, 50˚C for 45 s, and 72˚C for 30 s, and a final extension at 72˚C for 5 min. We used nuclease-free water as negative a control and DNA of *Dirofilaria immitis* as positive control.

We purified obtained PCR products using the E.Z.N.A.® Cycle Pure Kit (OMEGA Bio-Tek Inc., Norcross, GA, USA) according to the manufacturer's instructions. We aligned generated sequences and compared them to a variety of homologous *Onchocerca* sequences, including those reported in North America [4, 6, 22–24], in the nucleotide sequence database at National Center of Biotechnology Information (NCBI) using MEGA X 10.1 [25]. We conducted phylogenetic analysis using a maximum likelihood method and a General Time Reversible best fit model with gamma distribution (2,000 bootstrap replicates) and *Dirofilaria immitis* served as the species outgroup. All newly sequenced samples were submitted and accessioned in Gen-Bank (OP265723-39).

## Results

### Molt and sores

We compiled 322 observations of 15 individual moose (2015: n = 11; 2016: n = 12; 2021: n = 12) during molt. The first sign of molt starting was on May 5 and all moose completed molt by August 1 (Fig 2A). The first observation of a moose completing molt was on June 29 and the last was on July 25. Sore appearance was in coordination with molt; the first sore was observed June 9, 35 days after the start of molt (Fig 2). All moose had sores by July 5, and continued to have sores through summer (Fig 2B). In 2021, moose reached an average of 12 sores per hind leg per individual from July 22 to August 9, ranging from 3 to 25 sores (Fig 3). The first evidence of sores starting to heal was on July 30, and all moose had some evidence of sores starting to heal by August 10. All moose had sores that remained open through August.

### Thermal analysis

A total of 93 images from 12 moose were used for analysis. Maximum hind leg temperature was correlated with Julian day and ambient air temperature ($R^2$ [2df] = 0.2485, $P < 0.001$; Fig 4). Similarly, minimum leg temperature was correlated with Julian day and ambient air temperature ($R^2$ [2df] = 0.6128, $P < 0.001$; Fig 4). Maximum perianal temperature was correlated with Julian day and ambient air temperature as well ($R^2$ [2df] = 0.2489, $P < 0.001$; Fig 4). Maximum perianal and maximum hind leg temperature were consistently correlated with similar values across Julian days, unlike minimum hind leg temperature which increased across the season (Fig 4). Increasing temperature coincided with molt, the loss of hair, and the appearance of sores (Figs 2 and 4).

### Tissue samples and histology

Biopsies were taken from eight MRC moose from May 12 to June 6, 2021 prior to the appearance of sores. In two of the animals, there was a mild eosinophilic and lymphocytic dermatitis (Fig 5A). The skin biopsies from the other six animals were without significant pathologic findings. The sores of 10 MRC moose were biopsied from July 20 to July 22, 2021. In the skin biopsies from all 10 animals, there were similar histologic lesions. On the surface there were

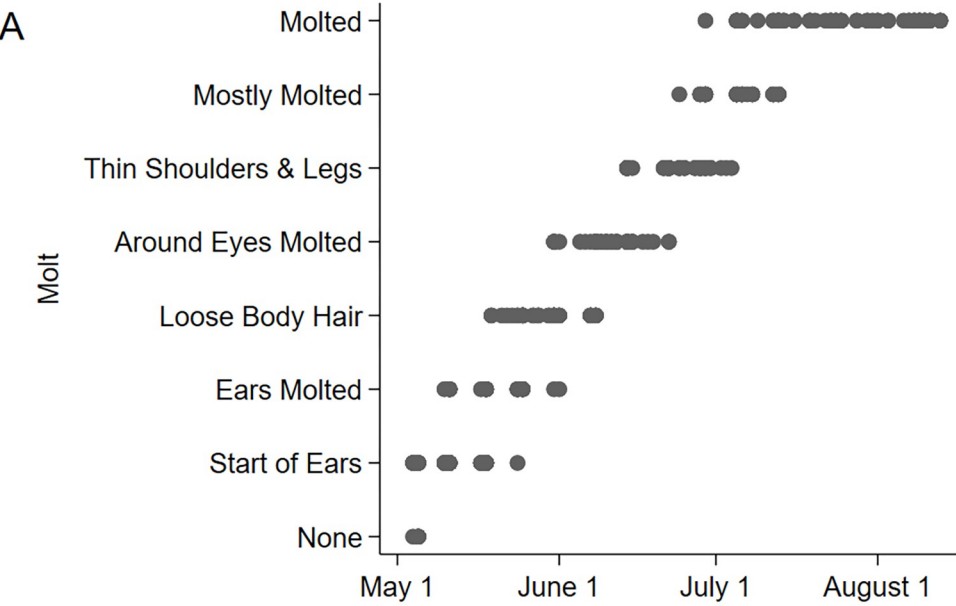

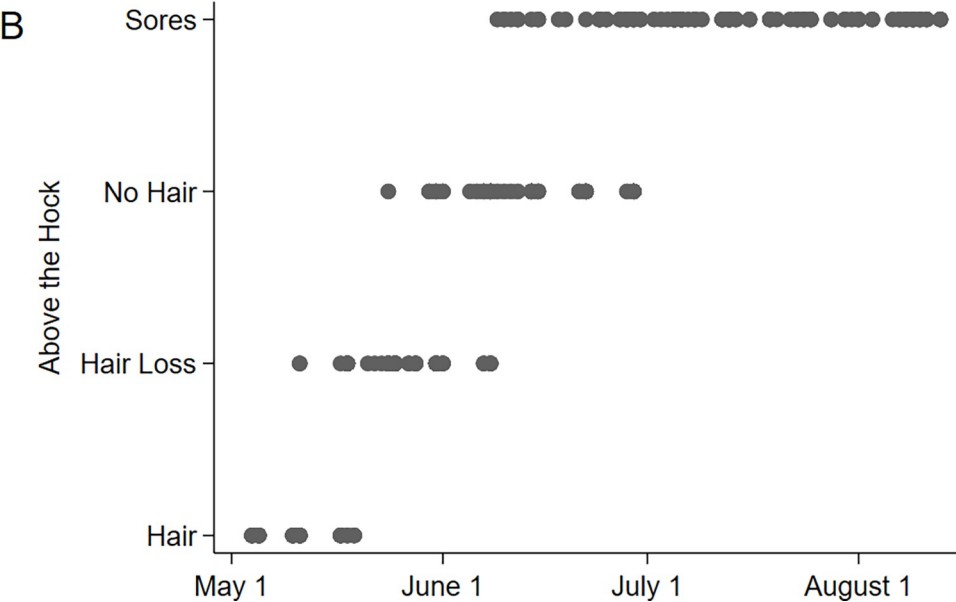

**Fig 2. Molt and sores.** A. The progression from winter coat to molted, summer coat in adult female moose (2015: n = 11; 2016: n = 12; 2021: n = 12). B. Molt of the hind leg area above the hocks and the appearance of sores. Points represent individual observations.

multifocally extensive serocellular crusts. The epidermis was frequently ulcerated. In the dermis there was a diffuse infiltration with many eosinophils (severe, multifocal, chronic, ulcerative and eosinophilic dermatitis; Fig 5B). However, no microfilariae were visible in the skin samples from live moose at the MRC. Tissue samples were collected from eight wild dead adult moose and one wild calf from the Kenai Peninsula from June 30 to July 23, 2021. In all of the adult animals, there were similar lesions as observed in the MRC moose with severe, ulcerative and eosinophilic dermatitis. In addition, there were multifocally large intraepidermal

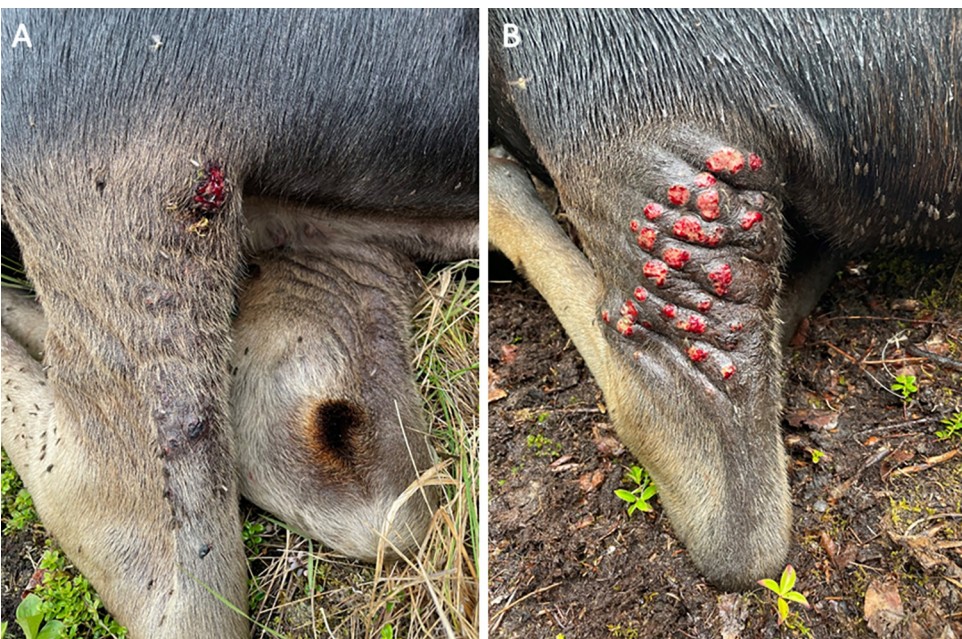

**Fig 3. Sores.** Variation in sores on the hind leg of adult female moose. A. Two sores (July 21, 2021). B. 25 sores (July 22, 2021).

pustules filled with eosinophils (Fig 5C). In four of the dead animals, there were microfilariae within intraepidermal pustules surrounded by degenerated eosinophils as an eosinophilic sleeve (Fig 5D).

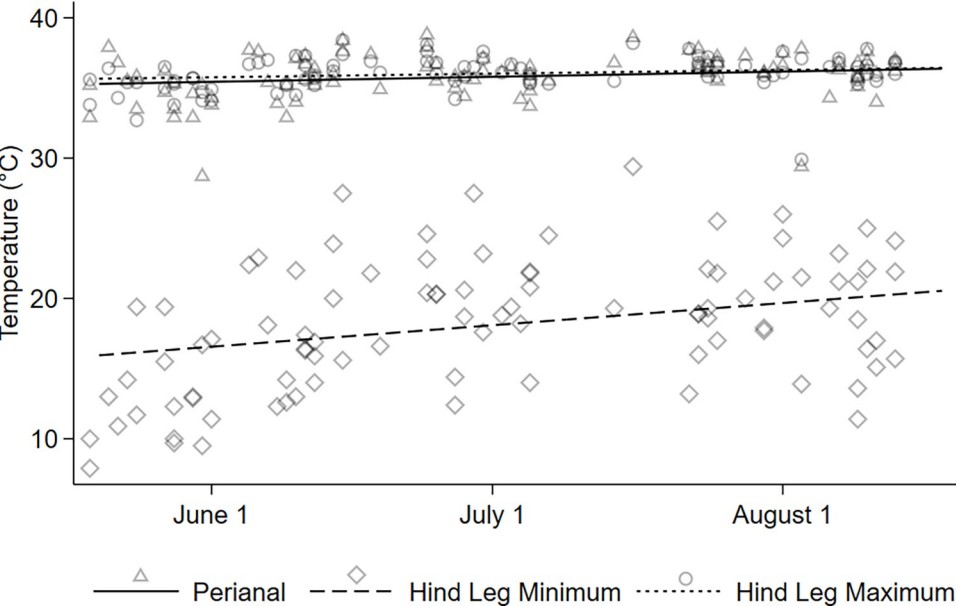

**Fig 4. Observed skin temperature (open symbols) in regions of interest on the caudal surface of adult female moose (n = 12) over summer.** Lines are predicted relationships between skin temperature and day of year for the temperature of the perianal (solid line and open triangles), hind leg maximum (dotted line and open circles), and hind leg minimum (dashed line and open diamonds).

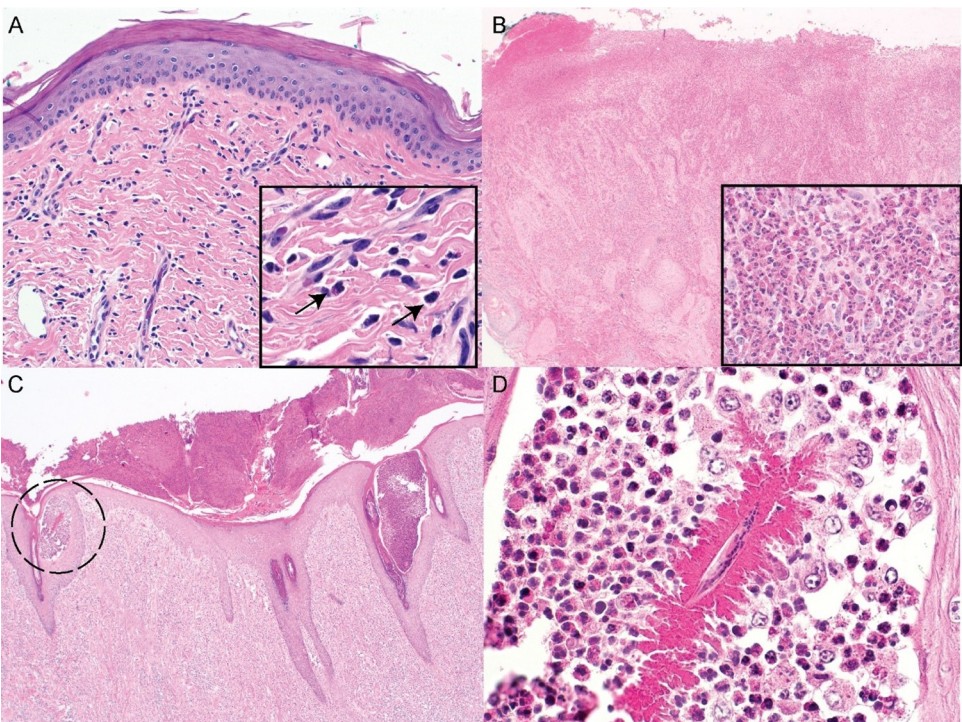

**Fig 5. Representative histologic photomicrographs of moose skin collected between May and July 2021.** A. Skin biopsy taken before appearance of sores in May, 2021. There are small numbers of perivascular eosinophils and lymphocytes in the dermis. No pustules or microfilariae are observed. Hematoxylin and eosin, 200x magnification; insert: higher magnification of small numbers of eosinophils (black arrows) and lymphocytes. Hematoxylin and eosin, 600x magnification. B. Biopsy from a leg sore of a live adult female moose. There is a diffuse, severe infiltration with eosinophils throughout the dermis. Note the diffuse ulceration and a serocellular crust on the left. Hematoxylin and eosin, 40x magnification; insert: higher magnification of the eosinophilic infiltration. Hematoxylin and eosin, 600x magnification. C. Skin sample from a dead wild moose sampled in July, 2021. There is a diffuse, severe infiltration with eosinophils throughout the dermis. Note multiple intraepidermal pustules (dashed circle). D. Higher magnification of the pustule highlighted in C. The pustule is filled with eosinophils with a microfilaria in the center surrounded by degenerated eosinophils forming an eosinophilic sleeve. Hematoxylin and eosin, 600x magnification.

## Genomic DNA

Out of the 26 samples processed, 17 produced a band from gel electrophoresis. Out of these samples 2/4 (50%) were from MRC moose taken before July, 4/9 (44.44%) were from MRC moose taken in July, and 7/8 (87.50%) were from wild moose taken between July and August, 2021. The remaining positives came from wild moose in 2020 (3/3; 100%) and samples from the same moose but different lesions (1/2; 50%). The sequences (396bp) were blasted using BLASTn tool to analyze their similarity with other published sequences available in online databases and it revealed very little similarity to other filarioid species. The closest parasites were *Onchocerca*, specifically *O. gutturosa* (94.59%), and decreased from there on. In addition, all 17 sequences had little intraspecies diversity (average pairwise identity of 99.85% with range of 100.00–98.96%) and likely belong to one *Onchocerca* species. Phylogenetic analysis revealed no specific group that clusters with the newly sequenced filarioid nematode DNA and it creates one distinct clade (Fig 6).

## Discussion

Morphological barriers such as skin and hair are the first line of defense used by ungulates to resist flies [1]. Moose calves do not develop sores, likely because their skin is covered by hair

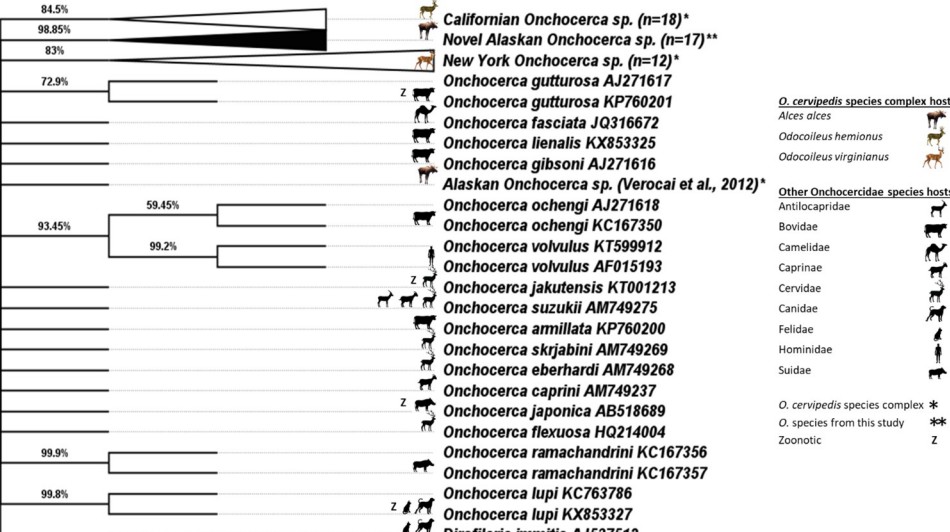

**Fig 6. Phylogenetic tree.** Phylogenetic tree created using a maximum likelihood method and a General Time Reversible best fit model with gamma distribution (2,000 bootstrap replicates) showing the relationship of all known *Onchocerca cervipedi*s species complex species in conjunction with other Onchocercidae species. At this time, all *Onchocerca* species comprised within the *O. cervipedi*s species complex (n = 4, denoted with *) form distinct clades including the positive samples (n = -17) from this study (denoted in black with **). These samples were accessioned in GenBank (OP265723-39).

throughout the summer [26]. Calves molt from a fuzzy natal coat to a winter coat that covered the majority of their body. Any area of the body exposed is heavily attacked by flies; the anus, around the eyes, and any injury resulting in a lack of hair. In adult moose, skin was exposed over the course of the molt from May to July (Fig 2). Fly numbers increased at the onset of molt, with the greatest numbers of flies on moose in July and August [26]. As molt was achieved in June and July, sores began to appear on the moose (Fig 2), which suggests that the flies are associated with the development of the sores. While there was a range in number of sores (3 to 25 sores per leg per individual), all adult moose had sores in July. Peak sore numbers (end of July and beginning of August) coincided with peak fly season [26] (Figs 2 and 3).

Hair as a physical morphological barrier and timing of the damage to the hind legs of the adult moose was further shown in the form of heat emitted, as captured by the thermal images, with the minimum temperature approaching perianal temperature and continuing to increase throughout the season (Fig 4). The perianal is a part of the moose that is hairless, an approach of minimum hind leg temperature towards perianal temperature shows the loss of a barrier (Fig 4). The perianal temperature (mean 35.8˚C) and maximum leg temperature (mean 36.0˚C) were slightly lower than summer core body temperature of moose (38.2˚C [27]). The initial rise in minimum hind leg temperature occurred with the loss of hair, and continued through the summer, coinciding with the breaks in the integument and abundance of sores, showing warming of the skin surrounding the sores and inflammation (Figs 1 and 4).

Eosinophilic dermatitis usually is a sign of allergic dermatitis/hypersensitivity, which is commonly associated with ectoparasites in wild animals. Twenty-five percent of the moose (2 of 8) had eosinophilic dermatitis before the sores developed, indicating that these animals may have been more susceptible and reacted earlier to the impact of flies than others. Additionally, *Onchocerca* DNA was found in 50% (2 of 4) of samples from moose without open sores. However, we cannot exclude hypersensitivity due to other environmental causes [3]. Interestingly, every single animal (10 of 10 moose) had developed severe eosinophilic and ulcerative

dermatitis ten weeks later. It is unclear if the severe eosinophilic infiltration is due to skin-swelling nematodes/microfilariae, or if the eosinophils are associated with a more severe reaction to ectoparasites. Even though no microfilariae were found histologically in these live animals (Fig 5), filarioid nematode DNA was found by genomic DNA extraction. *Onchocerca* DNA was found in 68% (14 of 21) of samples taken from open sores, 4 of which were from live MRC moose. Histology may therefore not be the most sensitive method to detect pathogens and is largely dependent on the sampling of the right location. For example, biopsies from horses with cutaneous habronemiasis often do not show nematodes in histological evaluations, despite their clear causation. The samples from the dead wild moose were much larger and more numerous per individual than the live moose biopsies, which increased the likelihood of finding microfilaria in histopathology.

Genetic analysis revealed a previously unknown genetic lineage within the genus *Onchocerca*, which is distinct from other genetic lineages with available COI sequence data, including those reported from North America (Fig 6). We use the terminology 'previously unknown genetic lineage' due to a lack of clustering with any specific group in online databases, and clustering into one distinct clade. Further work would need to be done to determine if the species is newly evolved, or just newly sequenced. The need for genetic evaluation of *Onchocerca* beyond morphological evaluation makes identification sparse. The serendipitous finding of a previously unknown genetic lineage of *Onchocerca* provides further evidence that, prior to the last decade of research, diversity of *Onchocerca* associated with North American ungulates was largely underestimated. At this time, *O. cervipedis* is no longer viewed as a single species, but rather a species complex with at least three to four genetic isolates, infecting at least three different ungulate species, with two of these distinct lineages infecting moose from northwestern North America [4, 6, 22]. It remains unclear which of these genetic lineages characterized to date correspond to the originally described *O. cervipedis*, which was based on specimens from white-tailed and mule deer [6, 22, 23]. Until recently, all reports of *Onchocerca* in various North American ungulates were assumed to involve *O. cervipedis*.

This previously uncharacterized genetic lineage of *Onchocerca* is the likely cause of the sores on the hind legs of moose, probably transmitted by black flies. This was a surprising finding as *O. cervipedis* predominately affects the subcutaneous tissue in the lower forelegs and below the hock in the hind legs of moose [4, 7], whereas the previously uncharacterized *Onchocerca* species seem to induce lesions further proximal on the legs. The impact of *Onchocerca* in moose is largely unknown, with findings of *Onchocerca* species in healthy moose and reports of only localized inflammatory reactions [4]. Black flies have been identified as biological vectors for various *Onchocerca* species, and have been observed feeding on moose [4, 7, 26]. Other species of *Onchocerca* use biting midges (Culicoides: Ceratopogonidae) as biological vectors. We observed that the majority of flies congregated on the hind end of the moose, from their rump to their hocks, and fed predominately in the areas that sores were observed. Further research is needed to determine which dipterans serve as vector for this newly characterized *Onchocerca*.

Sores and the previously uncharacterized species of *Onchocerca* were found in both MRC moose and wild moose from the Kenai Peninsula, showing the prevalence throughout the area. The impact of *Onchocerca* infection on individual moose health or moose populations has never to our knowledge been studied in detail. However, the location of infections and lesions associated with parasitism have been suggested to make infected animal more prone to predation [4]. Moose are an important part of the diet of many Alaskans, particularly rural Alaska where 60% of households harvest wild game and 86% consume wild game [28]. Therefore, concerns around food safety and security, including those associated with parasitic diseases deserve attention.

In conclusion, we found that the sores developing on the proximal aspect of the hind limbs of moose in the summer months are caused by a previously uncharacterized *Onchocerca* species. We correlated the emergence of sores with molt, and a lack of a barrier of protection in adult animals. These breaks in integument leave an ungulate exposed to pathogens, making them vulnerable, and challenging their health and fitness.

## Supporting information

**S1 Fig. Representative adult molt score photographs.** Photographs showing the eight stages of molt scores. Numbers correlated to molt scores in Table 1.
(TIF)

**S2 Fig. Representative adult hock score photographs.** Photographs showing the four stages of hock scores. Numbers correlated to hock scores in Table 1.
(TIF)

## Acknowledgments

We thank Nicholas Fowler for his tireless efforts in retrieving the wild moose samples.

## Author Contributions

**Conceptualization:** Bridgett M. Benedict, Perry S. Barboza, Dominique J. Wiener.

**Data curation:** Bridgett M. Benedict, Dominique J. Wiener.

**Formal analysis:** Bridgett M. Benedict, Perry S. Barboza, John A. Crouse, Katia R. Groch, Matthew R. Kulpa, Dan P. Thompson, Guilherme G. Verocai, Dominique J. Wiener.

**Funding acquisition:** Bridgett M. Benedict, Perry S. Barboza, John A. Crouse, Dominique J. Wiener.

**Investigation:** Bridgett M. Benedict, Perry S. Barboza, John A. Crouse, Matthew R. Kulpa, Dan P. Thompson, Guilherme G. Verocai, Dominique J. Wiener.

**Methodology:** Bridgett M. Benedict, Perry S. Barboza, John A. Crouse, Dan P. Thompson, Dominique J. Wiener.

**Project administration:** Bridgett M. Benedict, Perry S. Barboza, John A. Crouse, Dan P. Thompson, Dominique J. Wiener.

**Resources:** Bridgett M. Benedict, Perry S. Barboza, John A. Crouse, Dan P. Thompson, Dominique J. Wiener.

**Software:** Bridgett M. Benedict, Perry S. Barboza, Dominique J. Wiener.

**Supervision:** Bridgett M. Benedict, Perry S. Barboza, John A. Crouse, Dan P. Thompson, Dominique J. Wiener.

**Validation:** Bridgett M. Benedict, Perry S. Barboza, Dominique J. Wiener.

**Visualization:** Bridgett M. Benedict, Perry S. Barboza, Dominique J. Wiener.

**Writing – original draft:** Bridgett M. Benedict, Matthew R. Kulpa, Guilherme G. Verocai, Dominique J. Wiener.

**Writing – review & editing:** Bridgett M. Benedict, Perry S. Barboza, John A. Crouse, Matthew R. Kulpa, Dan P. Thompson, Guilherme G. Verocai, Dominique J. Wiener.

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
