## [Decision Letter · Decision Letter 0]

17 Oct 2022

PONE-D-22-25378Sores of boreal moose reveal a novel species of parasitic nematode within the genus OnchocercaPLOS ONE

Dear Dr. Benedict,

Thank you for submitting your manuscript to PLOS ONE. After careful consideration, we feel that it has merit but does not fully meet PLOS ONE’s publication criteria as it currently stands. Therefore, we invite you to submit a revised version of the manuscript that addresses the points raised during the review process.

I appreciate the effort of the authors in compiling this effort on identifying a novel Onchocerca sp. in moose. While I understand why the stages of molt are important in adult moose, I am not sure that level of detail is necessary in text or supplemental figures. Certainly not in moose calves considering none were detected in moose calves. I would suggest the authors spend considerably more text and effort in addressing comments from Reviewer 2 in expanding the methods and results to improve clarity on methods of scoring and whether that is as valuable as the histology and genetic analysis. As indicated by the reviewers, what is the basis for your molt scoring and is it a published scoring system or developed by the authors? If the authors are proposing a molt scoring system then that must be an objective and presented as such throughout the manuscript instead of focusing on adults vs. calves, captive vs. wild, etc.

All of this could be set up much better in the final paragraph in the introduction by explicitly stating your objectives. It appears the authors have 3 objectives but Objective 2 (calves) is simply molt progression which could be removed entirely? The very vague objectives could then focus less on molt patterns (unless it is being proposed here) and more on the findings and identification of the novel Onchocerca sp. as suggested by Reviewer 2. Please submit your revised manuscript by 21 November 2022. If you will need more time than this to complete your revisions, please reply to this message or contact the journal office at plosone@plos.org. Please include the following items when submitting your revised manuscript:A rebuttal letter that responds to each point raised by the academic editor and reviewer(s). You should upload this letter as a separate file labeled 'Response to Reviewers'.A marked-up copy of your manuscript that highlights changes made to the original version. You should upload this as a separate file labeled 'Revised Manuscript with Track Changes'.An unmarked version of your revised paper without tracked changes. You should upload this as a separate file labeled 'Manuscript'.

We look forward to receiving your revised manuscript.

Kind regards,

W. David Walter, Ph.D.

Academic Editor

PLOS ONE

Journal Requirements:

3. We note that Figures 3, S1, S2 and S3 in your submission contain copyrighted images. All PLOS content is published under the Creative Commons Attribution License (CC BY 4.0), which means that the manuscript, images, and Supporting Information files will be freely available online, and any third party is permitted to access, download, copy, distribute, and use these materials in any way, even commercially, with proper attribution. For more information, see our copyright guidelines: http://journals.plos.org/plosone/s/licenses-and-copyright.

a. You may seek permission from the original copyright holder of Figures 3, S1, S2 and S3  to publish the content specifically under the CC BY 4.0 license. 

Additional Editor Comments:

Dates and formatting of dates throughout the manuscript are not consistent as written and very confusing.

Captive versus wild moose were described in the same paragraph of the Material and Methods but should not be treated as if all methods were done on both captive and wild. Why not organize Materials and Methods by the objective? Clearly, molt pattern and thermal imagery was not done on wild moose and cutting out the sore was not done in captive moose. The authors then go back and forth on Dates of molting and collecting roadkilled moose.

Lines 236-239: Again, it is not clear what Methods and Results of captive calf molt progression adds to this manuscript?

Figure caption text should be in consistent formatting. For example, panels are referenced with (A) and B. in Figure 2, A. in Figure 3, and A: in Figure 5.

Reviewers' comments:

Reviewer's Responses to Questions

**Comments to the Author**

1. Is the manuscript technically sound, and do the data support the conclusions?

Reviewer #1: Yes

Reviewer #2: Partly

2. Has the statistical analysis been performed appropriately and rigorously? 

Reviewer #1: N/A

Reviewer #2: I Don't Know

3. Have the authors made all data underlying the findings in their manuscript fully available?

Reviewer #1: Yes

Reviewer #2: Yes

4. Is the manuscript presented in an intelligible fashion and written in standard English?

Reviewer #1: Yes

Reviewer #2: Yes

5. Review Comments to the Author

Reviewer #1: This paper is a detailed account of a novel type of Onchocerca spp. retrieved from sores on moose legs. The methods were well-documented and the accompanying photographs and figures added a lot of value to the manuscript. There were a few places where I think clarification is warranted. Please see below. Overall, a terrific effort!

Abstract: please conclude with a statement about the significance of your findings...similar to what is stated in lines 58-370.

Line 67: suggested rephrasing: "Despite a superficial resemblance suggesting a similar cause, these sores were not cutaneous habronemiasis."

Line 74: please provide taxonomic information for "leeches"

Also, could leeches perhaps be responsible for sores if they were feeding on moose when moose were foraging in wetlands? The leeches may not be observable if the moose is standing in belly-deep water.

Line 80: suggest "whereas" in place of "while"

Line 85: is the degree of inflammation being evaluated via the thermal imagery? This brings up a general concern about how the thermal imagery is being used and interpreted.

Line 98: please provide taxonomic information for "black spruce"

Line 126: The entirety of the hind leg thermal imagery section is in passive voice. I would suggest rephrasing here (and elsewhere in the manuscript) that more active voice be incorporated.

Also, please say here why the thermal imagery is being used.

Line 194: This section is also very passive. Please think about rephrasing. Even if the work was not directly conducted by the authors, consider stating things like, "Technicians used nuclease-free water as a negative control..."

Using active voice would also help clarify the sentence in lines 195-196. I needed to reread the sentence a few times to understand what the subject of the sentence was.

Line 218: please indicate here that the observations were from 15 "unique" adult moose.

Line 247: Here you state there is a peak in temperature at the end of July, yet the trendline continues to increase. I am not seeing a peak. Can you clarify this?

Lines 295-298: This really cool interpretation belongs in the discussion, in my opinion.

First paragraph of discussion: Much of the information about blackfly phenology comes from a manuscript in review. Are there other sources in the literature that can be used?

Line 325: Again here, I don't see the peak.

Line 328: "Only a few of the moose..." but still, it is 25% of those sampled, so I don't think this should be discounted.

Line 354: Please check throughout the use of "black flies" vs. "blackflies" and standardize if appropriate.

Line 360: "never" is a strong word...maybe soften to "never to our knowledge"

Congratulations on the discovery of an uncharacterized Onchocerca species! I hope these comments prove useful!

Sincerely,

-William (Bill) Severud

Reviewer #2: Benedict et al. ‘Sores of boreal moose reveal a novel species of parasitic nematode within the genus Onchocerca.’

The authors used PCR and histology to describe lesions on the hind legs of moose and showed evidence of a genetically unique Onchocerca sp to be associated with lesions in adult moose. Molt patterns and thermal imaging were used to support the hypothesis that sores develop later in the year as hair is sparse during molt. The manuscript is reasonably well-written and provides useful and interesting information.

Most of my suggestions are to expand on sections of the methods and results to improve clarity, as some methods of scoring were not very objective or well-described. Minor suggestions are also suggested at the bottom.

Terminology changed throughout the manuscript and became confusing to follow at times. Finally, the title implies that this is a novel species and while that may be true, a thorough discussion on the challenges of characterizing filarids is warranted considering isolation and morphological evaluation of these nematodes is challenging. The authors use ‘previously uncharacterized’, ‘novel operational taxonomic unit’ and ‘novel species’ throughout the text and title. Since these have different meanings and implications, I strongly suggest a more detailed discussion paragraph about what is explicitly unique about these nematodes, why the OTU terminology is used, and what else needs to be done to further emphasize if this is a ‘novel species.’ The terminology in the abstract seems the most appropriate: ‘previously unknown genetic lineage.’

Lines 106-109: Has this molt score been used in the literature or is this a novel scoring system? Is it based on known progression of molt? If so, include a reference similar to that in the calf molt paragraph.

Line 110: Without context, assessing the ‘level of inflammation’ grossly seems extremely subjective and, in my opinion, without much merit in the absence of histology. If ulceration, size of individual lesions, ‘swelling’ (as is sometimes used in the text), extent of hemorrhage or hyperkeratosis, etc. is what is being assessed, I suggest saying this specifically here in the methods or describing what is meant by ‘level of inflammation’ to the same detail as the molt score. When consistent terminology is defined, also consider rephrasing at lines 233 and 234. If ‘scoring’ inflammation is not important for statistics or other aspects of the results, you could also remove these modifiers completely.

Table 1: hock scores did not appear to be introduced in the text and lacks context in the methods. If line 110 is meant to refer to the level of ‘inflammation’ or meant to introduce the ‘hock score’, I recommend being more explicit.

Lines 295-298 and 345: See comment above regarding novel sequence, OTU, and species.

Results lines 285-289: It seems there are two goals here: 1: to compare PCR prevalence against time to support that later in the year with reduced hair cover, PCR prevalence increases, and 2: to show PCR prevalence in lesions vs non lesions. The first point seems to be appropriately addressed, but not the second point. A simple descriptive statistic on the percentage of ‘sores’ with onchocerca DNA present compared to the percentage of ‘normal tissue’ with onchocerca DNA present would be useful to provide more evidence of causation.

Other important discussion points not included:

1: why there is a lack of sores in other hairless areas of the body ‘eyes, perianal region, etc.’ particularly since flies are known to be present here

2: biopsies from horses with ‘summer sores’ often lack nematodes in section, yet we still assume they are the cause similar to nematodes in this study. This point will provide more support for causation.

Minor suggestions

Line 39: plural subject � ‘are important barriers between the animal…’

Line 46: ‘often have been documented with round sores….’

Lines 82-87: ‘tracked’ vs ‘track’, etc. to keep in past or present tense

Line 94-97: Are these moose captive or free ranging, since both terminology is used? The implications are very different, so I suggest being clearer in this section

Line 168: ‘Top’ of the leg is a little confusing; suggesting using ‘proximal’ or ‘lateral aspect of exposed leg’ or something similar for clarity

Line 192: 4 um ‘thick’ sections, for clarity

Line 219, 265: August 1st and June 30th, etc. to be consistent with how other dates are in the text

Line 227: a reminder of when the ‘end of the study’ occurred may be useful

Figure 4: could the symbols be included in the text figure key with the solid/dashed lines?

Figure 5B: Is the pale staining dermis necrosis or autolysis? If necrosis, that seems to be an important modifier here.

Line 308: While this is probably true, I don’t think you ‘confirmed’ that this is the reason that moose calves did not develop sores ,and I suggest hedging a little bit on this statement.

Line 289: I am unclear what is meant by ‘same moose skin but different lesions.’

6. PLOS authors have the option to publish the peer review history of their article (what does this mean?). If published, this will include your full peer review and any attached files.

Reviewer #1: **Yes: **William J Severud

Reviewer #2: No

---

## [Author Response · Author response to Decision Letter 0]

8 Nov 2022

Manuscript PONE-D-22-25378

Sores of boreal moose reveal a previously unknown genetic lineage of parasitic nematode within the genus Onchocerca

Answers to the editor and the reviewers

We are thankful for the editor and reviewers’ constructive comments that helped to improve and clarify the manuscript. In the following we address the comments individually. We also highlighted the changes in the manuscript by track changes.

Editor

I appreciate the effort of the authors in compiling this effort on identifying a novel Onchocerca sp. in moose. While I understand why the stages of molt are important in adult moose, I am not sure that level of detail is necessary in text or supplemental figures. Certainly not in moose calves considering none were detected in moose calves. I would suggest the authors spend considerably more text and effort in addressing comments from Reviewer 2 in expanding the methods and results to improve clarity on methods of scoring and whether that is as valuable as the histology and genetic analysis. As indicated by the reviewers, what is the basis for your molt scoring and is it a published scoring system or developed by the authors? If the authors are proposing a molt scoring system then that must be an objective and presented as such throughout the manuscript instead of focusing on adults vs. calves, captive vs. wild, etc. 

All of this could be set up much better in the final paragraph in the introduction by explicitly stating your objectives. It appears the authors have 3 objectives but Objective 2 (calves) is simply molt progression which could be removed entirely? The very vague objectives could then focus less on molt patterns (unless it is being proposed here) and more on the findings and identification of the novel Onchocerca sp. as suggested by Reviewer 2.

Response: We have removed calf molt as an objective, and from the body of the manuscript. We agree that the methods on calf molt were distracting and did not add to the manuscript. A few points remain in the discussion, noting the lack of sores found in calves and this relation to molt. The manuscript is now centered around adult moose; describing the progression of molt to sores and identifying the cause for the sores. We have clarified that we designed the adult molt and hock scoring system, but do not think it warrants a main objective point (lines 84-88; lines 106-107). We removed the scoring regarding the degree of inflammation because of to its subjectivity, as pointed out, and did not feel it added to the paper. 

Journal Requirements:

Response: The manuscript has been checked to meet PLOS ONE requirements.

Response: The datasets have been published at:

Benedict, Bridgett, 2022, "Moose Sores and Molt", https://doi.org/10.18738/T8/HGPOGJ, Texas Data Repository, V1

GenBank Accession No. OP265723-39

3. We note that Figures 3, S1, S2 and S3 in your submission contain copyrighted images. All PLOS content is published under the Creative Commons Attribution License (CC BY 4.0), which means that the manuscript, images, and Supporting Information files will be freely available online, and any third party is permitted to access, download, copy, distribute, and use these materials in any way, even commercially, with proper attribution. For more information, see our copyright guidelines: http://journals.plos.org/plosone/s/licenses-and-copyright.

Response: All images are owned by the authors of this manuscript and are approved for use. 

Additional Editor Comments:

Dates and formatting of dates throughout the manuscript are not consistent as written and very confusing.

Response: Dates have been made consistent throughout and are hopefully less confusing. 

Captive versus wild moose were described in the same paragraph of the Material and Methods but should not be treated as if all methods were done on both captive and wild. Why not organize Materials and Methods by the objective? Clearly, molt pattern and thermal imagery was not done on wild moose and cutting out the sore was not done in captive moose. The authors then go back and forth on Dates of molting and collecting roadkilled moose.

Response: Text on wild moose materials and methods has been moved to its own section and given the title “Wild moose sampling” (lines 173-183)

Lines 236-239: Again, it is not clear what Methods and Results of captive calf molt progression adds to this manuscript?

Response: All data and analysis on calf molt have been removed from the manuscript. 

Figure caption text should be in consistent formatting. For example, panels are referenced with (A) and B. in Figure 2, A. in Figure 3, and A: in Figure 5.

Response: A-D’s have been made consistent in Figure captions (lines 222-224; 258-270) 

Reviewer #1:

Comments to the Author

Reviewer #1: This paper is a detailed account of a novel type of Onchocerca spp. retrieved from sores on moose legs. The methods were well-documented and the accompanying photographs and figures added a lot of value to the manuscript. There were a few places where I think clarification is warranted. Please see below. Overall, a terrific effort!

Abstract: please conclude with a statement about the significance of your findings...similar to what is stated in lines 58-370.

Response: A new sentence has been added about the significance (lines 37-38).

Line 67: suggested rephrasing: "Despite a superficial resemblance suggesting a similar cause, these sores were not cutaneous habronemiasis."

Response: Sentence has been rephrased similarly to as suggested (lines 67-70).

Line 74: please provide taxonomic information for "leeches"

Response: Class added for leech (line 76).

Also, could leeches perhaps be responsible for sores if they were feeding on moose when moose were foraging in wetlands? The leeches may not be observable if the moose is standing in belly-deep water.

Response: That’s possible but I would have thought there would have been some observation of leeches being attached after the moose emerges from the water.

Line 80: suggest "whereas" in place of "while"

Response: While has been replaced with whereas (line 82). 

Line 85: is the degree of inflammation being evaluated via the thermal imagery? This brings up a general concern about how the thermal imagery is being used and interpreted.

Response: The degree of inflammation has been removed due to subjectivity. The thermal imagery was evaluated using thermal software and statistical analysis, this was not subjective and not used for degree of inflammation. See methods (lines 128-132; lines 139-148). 

Line 98: please provide taxonomic information for "black spruce"

Response: Taxonomic name for black spruce has been added (line 99).

Line 126: The entirety of the hind leg thermal imagery section is in passive voice. I would suggest rephrasing here (and elsewhere in the manuscript) that more active voice be incorporated.

Response: This section and the entire methods section has been changed to a more active voice.

Also, please say here why the thermal imagery is being used.

Response: The first sentence has been revised to explain why thermal imagery was used (lines 117-118). 

Line 194: This section is also very passive. Please think about rephrasing. Even if the work was not directly conducted by the authors, consider stating things like, "Technicians used nuclease-free water as a negative control..."

Using active voice would also help clarify the sentence in lines 195-196. I needed to reread the sentence a few times to understand what the subject of the sentence was.

Response: This section has been reworded into active voice.

Line 218: please indicate here that the observations were from 15 "unique" adult moose.

Response: Added “individual” moose (line 212).

Line 247: Here you state there is a peak in temperature at the end of July, yet the trendline continues to increase. I am not seeing a peak. Can you clarify this?

Response: Peaking has been changed to increasing (lines 235-236). 

Lines 295-298: This really cool interpretation belongs in the discussion, in my opinion.

Response: The sentence has been moved to the discussion (lines 326-328).

First paragraph of discussion: Much of the information about blackfly phenology comes from a manuscript in review. Are there other sources in the literature that can be used?

Response: The reference used is for a paper on the effects of flies on moose calves. There isn’t anything out to replace it but it should be published soon. 

Line 325: Again here, I don't see the peak.

Response: Sentences has been changed to say “increasing” instead of “peaking” (lines 301-304; lines 307-309)

Line 328: "Only a few of the moose..." but still, it is 25% of those sampled, so I don't think this should be discounted.

Response: The sentence has been changed from “only a few” to “25%” (lines 311-313).

Line 354: Please check throughout the use of "black flies" vs. "blackflies" and standardize if appropriate.

Response: Blackflies has been changed to two words (line 346).

Line 360: "never" is a strong word...maybe soften to "never to our knowledge"

Response: Phrase has been added (line 355).

Congratulations on the discovery of an uncharacterized Onchocerca species! I hope these comments prove useful!

Sincerely,

-William (Bill) Severud

Response: Thank you! Yes, the comments were very useful and were all used to edit the manuscript. The largest edits were the removal of the degree of inflammation and rewording the thermal analysis results. 

Reviewer #2:

Reviewer #2: Benedict et al. ‘Sores of boreal moose reveal a novel species of parasitic nematode within the genus Onchocerca.’

The authors used PCR and histology to describe lesions on the hind legs of moose and showed evidence of a genetically unique Onchocerca sp to be associated with lesions in adult moose. Molt patterns and thermal imaging were used to support the hypothesis that sores develop later in the year as hair is sparse during molt. The manuscript is reasonably well-written and provides useful and interesting information.

Most of my suggestions are to expand on sections of the methods and results to improve clarity, as some methods of scoring were not very objective or well-described. Minor suggestions are also suggested at the bottom.

Terminology changed throughout the manuscript and became confusing to follow at times. Finally, the title implies that this is a novel species and while that may be true, a thorough discussion on the challenges of characterizing filarids is warranted considering isolation and morphological evaluation of these nematodes is challenging. The authors use ‘previously uncharacterized’, ‘novel operational taxonomic unit’ and ‘novel species’ throughout the text and title. Since these have different meanings and implications, I strongly suggest a more detailed discussion paragraph about what is explicitly unique about these nematodes, why the OTU terminology is used, and what else needs to be done to further emphasize if this is a ‘novel species.’ The terminology in the abstract seems the most appropriate: ‘previously unknown genetic lineage.’

Response: The calf molt sections have been removed and the further explanation has been added to the adult molt scoring. 

We have changed the terminology to “previously unknown genetic lineage” throughout and have added a paragraph explaining this choice of wording and explained the need for genetic identification of these species (lines 104-107; lines 318-322; lines 326-332). 

Lines 106-109: Has this molt score been used in the literature or is this a novel scoring system? Is it based on known progression of molt? If so, include a reference similar to that in the calf molt paragraph.

Response: To our knowledge, there isn’t a previously used molt scoring system for moose. This was something that we designed to best capture what we had seen occurring in previous years and during our observations. A sentence has been added to clarify (lines 106-107).

Line 110: Without context, assessing the ‘level of inflammation’ grossly seems extremely subjective and, in my opinion, without much merit in the absence of histology. If ulceration, size of individual lesions, ‘swelling’ (as is sometimes used in the text), extent of hemorrhage or hyperkeratosis, etc. is what is being assessed, I suggest saying this specifically here in the methods or describing what is meant by ‘level of inflammation’ to the same detail as the molt score. When consistent terminology is defined, also consider rephrasing at lines 233 and 234. If ‘scoring’ inflammation is not important for statistics or other aspects of the results, you could also remove these modifiers completely.

Response: Degree of inflammation has been removed from the manuscript entirely due to being vague and subjective. 

Table 1: hock scores did not appear to be introduced in the text and lacks context in the methods. If line 110 is meant to refer to the level of ‘inflammation’ or meant to introduce the ‘hock score’, I recommend being more explicit.

Response: Degree of inflammation has been removed and text has been modified to describe hock scoring (lines 104-106).

Lines 295-298 and 345: See comment above regarding novel sequence, OTU, and species.

Response: This sentence has been moved to the discussion and wording has been changed throughout to “previously unknown”.

Results lines 285-289: It seems there are two goals here: 1: to compare PCR prevalence against time to support that later in the year with reduced hair cover, PCR prevalence increases, and 2: to show PCR prevalence in lesions vs non lesions. The first point seems to be appropriately addressed, but not the second point. A simple descriptive statistic on the percentage of ‘sores’ with onchocerca DNA present compared to the percentage of ‘normal tissue’ with onchocerca DNA present would be useful to provide more evidence of causation.

Response: Discussion has been expanded to include percentages of what was found and the increase in prevalence with sores (lines 311-314; lines 320-321).

Other important discussion points not included:

1: why there is a lack of sores in other hairless areas of the body ‘eyes, perianal region, etc.’ particularly since flies are known to be present here

Response: We can’t say for sure why the eyes and perianal do not have sores. But we can say that we observed the majority of flies on the area of the moose with sores, and observed them feeding. A sentence as been added to clarify this (lines 349-351).

2: biopsies from horses with ‘summer sores’ often lack nematodes in section, yet we still assume they are the cause similar to nematodes in this study. This point will provide more support for causation.

Response: That’s a good point, and has been added (lines 322-324).

Minor suggestions

Line 39: plural subject � ‘are important barriers between the animal…’

Response: Subject made singular (line 41).

Line 46: ‘often have been documented with round sores….’

Response: Sentence has reworded (line 48).

Lines 82-87: ‘tracked’ vs ‘track’, etc. to keep in past or present tense

Response: Sentence has been removed due to other comments.

Line 94-97: Are these moose captive or free ranging, since both terminology is used? The implications are very different, so I suggest being clearer in this section

Response: “Free ranging” has been removed.

Line 168: ‘Top’ of the leg is a little confusing; suggesting using ‘proximal’ or ‘lateral aspect of exposed leg’ or something similar for clarity

Response: “Top” changed to “proximal” (line 159).

Line 192: 4 um ‘thick’ sections, for clarity

Response: Thick was added (line 186).

Line 219, 265: August 1st and June 30th, etc. to be consistent with how other dates are in the text

Response: “th” was removed throughout.

Line 227: a reminder of when the ‘end of the study’ occurred may be useful

Response: Sentences revised to say “through August” (line 220).

Figure 4: could the symbols be included in the text figure key with the solid/dashed lines?

Response: Figure modified to include symbols on key. 

Figure 5B: Is the pale staining dermis necrosis or autolysis? If necrosis, that seems to be an important modifier here.

Response: The pale staining is dermis, there is no necrosis or autolysis. The dermis only seems paler than usual as it is next to the abundant eosinophils that have a more intense eosinophilic staining.

Line 308: While this is probably true, I don’t think you ‘confirmed’ that this is the reason that moose calves did not develop sores ,and I suggest hedging a little bit on this statement.

Response: The moose calf molt sections have been removed based on other comments, and this sentence has been modified (lines 292-293).

Line 289: I am unclear what is meant by ‘same moose skin but different lesions.’

Response: “Skin” removed to clarify sentence (lines 275-276).

---

## [Decision Letter · Decision Letter 1]

22 Nov 2022

PONE-D-22-25378R1Sores of boreal moose reveal a previously unknown genetic lineage of parasitic nematode within the genus OnchocercaPLOS ONE

Dear Dr. Benedict,

Thank you for submitting your manuscript to PLOS ONE. After careful consideration, we feel that it has merit but does not fully meet PLOS ONE’s publication criteria as it currently stands. Therefore, we invite you to submit a revised version of the manuscript that addresses the points raised during the review process.

The reviewers acknowledged a great improvement of the manuscript. It can be accepted following some small edits and clarifications, as suggested by Reviewer 2.

We look forward to receiving your revised manuscript.

Kind regards,

Angela Monica Ionica, Ph.D.

Academic Editor

PLOS ONE

Journal Requirements:

Reviewers' comments:

Reviewer's Responses to Questions

**Comments to the Author**

1. If the authors have adequately addressed your comments raised in a previous round of review and you feel that this manuscript is now acceptable for publication, you may indicate that here to bypass the “Comments to the Author” section, enter your conflict of interest statement in the “Confidential to Editor” section, and submit your "Accept" recommendation.

Reviewer #1: All comments have been addressed

Reviewer #2: (No Response)

2. Is the manuscript technically sound, and do the data support the conclusions?

Reviewer #1: (No Response)

Reviewer #2: Yes

3. Has the statistical analysis been performed appropriately and rigorously? 

Reviewer #1: (No Response)

Reviewer #2: Yes

4. Have the authors made all data underlying the findings in their manuscript fully available?

Reviewer #1: (No Response)

Reviewer #2: Yes

5. Is the manuscript presented in an intelligible fashion and written in standard English?

Reviewer #1: (No Response)

Reviewer #2: Yes

6. Review Comments to the Author

Reviewer #1: (No Response)

Reviewer #2: Benedict et al. sores of boreal moose....

The authors have drastically improved the manuscript. I have a few minor suggestions below to help improve clarity of some sentences. Nice study with interesting findings.

Track changed version; 114-116; new sentence isn’t clear to me. Perhaps it’s the use of ‘what’ in line 115?

Track changed version line 137; can thermal imagery capture inflammation? Perhaps ‘ to track hair loss as a possible indicator of inflammation’?

Track changed version line 204-207; do these sentences say the same thing?

Track changed version line 338; need complete sentence after semicolon

7. PLOS authors have the option to publish the peer review history of their article (what does this mean?). If published, this will include your full peer review and any attached files.

Reviewer #1: **Yes: **William J Severud

Reviewer #2: No

---

## [Author Response · Author response to Decision Letter 1]

23 Nov 2022

Journal Requirements

Response: All citations have been check for accuracy and none have been retracted. 

Reviewers' Comments

Reviewer #2: Benedict et al. sores of boreal moose....

The authors have drastically improved the manuscript. I have a few minor suggestions below to help improve clarity of some sentences. Nice study with interesting findings.

Response: Thank you for your help and guidance! We appreciate your attention to detail in catching these typos and grammatical error, and have corrected them accordingly. 

Track changed version; 114-116; new sentence isn’t clear to me. Perhaps it’s the use of ‘what’ in line 115?

Response: The typo has been corrected; ‘what’ has been changed to ‘that’ (line 106).

Track changed version line 137; can thermal imagery capture inflammation? Perhaps ‘ to track hair loss as a possible indicator of inflammation’?

Response: It would be hard to decipher warming due to just loss of hair versus warming due to inflammation; ‘and inflammation’ has been removed (lines 117-118).

Track changed version line 204-207; do these sentences say the same thing?

Response: Yes, the sentences have been combined (lines 174-176).

Track changed version line 338; need complete sentence after semicolon

Response: The sentence has been made complete (lines 293-294).

---

## [Editor Report · Decision Letter 2]

28 Nov 2022

Sores of boreal moose reveal a previously unknown genetic lineage of parasitic nematode within the genus Onchocerca

PONE-D-22-25378R2

Dear Dr. Benedict,

We’re pleased to inform you that your manuscript has been judged scientifically suitable for publication and will be formally accepted for publication once it meets all outstanding technical requirements.

Kind regards,

Angela Monica Ionica, Ph.D.

Academic Editor

PLOS ONE
---

## [Editor Report · Acceptance letter]

2 Dec 2022

PONE-D-22-25378R2 

Sores of boreal moose reveal a previously unknown genetic lineage of parasitic nematode within the genus *Onchocerca*

Dear Dr. Benedict:

I'm pleased to inform you that your manuscript has been deemed suitable for publication in PLOS ONE. Congratulations! Your manuscript is now with our production department. 

Kind regards, 

on behalf of

Dr. Angela Monica Ionica 

Academic Editor

PLOS ONE